# Polymorphisms in the hypoxia inducible factor binding site of the macrophage migration inhibitory factor gene promoter in schizophrenia

**Satoshi Okazaki**[1], **Shuken Boku**[1,2]*, **Yuichiro Watanabe**[3], **Ikuo Otsuka**[1], **Tadasu Horai**[1], **Ryo Morikawa**[3], **Atsushi Kimura**[1], **Naofumi Shimmyo**[1], **Takaki Tanifuji**[1], **Toshiyuki Someya**[3], **Akitoyo Hishimoto**[1,4]

**1** Department of Psychiatry, Kobe University Graduate School of Medicine, Kobe, Japan, **2** Department of Neuropsychiatry, Kumamoto University Faculty of Life Sciences, Kumamoto, Japan, **3** Department of Psychiatry, Niigata University Graduate School of Medical and Dental Sciences, Niigata, Japan, **4** Department of Psychiatry, Yokohama City University Graduate School of Medicine, Yokohama, Japan

* boku.shuken@kuh.kumamoto-u.ac.jp

**Data Availability Statement:** All relevant data are within the paper and its Supporting Information files.

## Abstract

### Background

Macrophage migration inhibitory factor (MIF) is a multifunctional cytokine that promotes neurogenesis and neuroprotection. MIF is predominantly expressed in astrocytes in the brain. The serum MIF level and microsatellites/single nucleotide polymorphisms (SNPs) in the *MIF* gene promoter region are known to be associated with schizophrenia (SCZ). Interestingly, previous studies reported that hypoxia, an environmental risk factor for SCZ, induced MIF expression through binding of the hypoxia inducible factor (HIF)-1 to the hypoxia response element (HRE) in the *MIF* promoter.

### Methods

We investigated the involvement of MIF in SCZ while focusing on the HIF pathway. First, we conducted an association study of the SNP rs17004038 (C>A) in the HRE of the *MIF* promoter between 1758 patients with SCZ and 1507 controls. Next, we investigated the effect of hypoxia on MIF expression in primary cultured astrocytes derived from neonatal mice forebrain.

### Results

SNP rs17004038 was significantly associated with SCZ ($p = 0.0424$, odds ratio = 1.445), indicating that this SNP in the HRE of the *MIF* promoter was a genetic risk factor for SCZ. Hypoxia induced *MIF* mRNA expression and MIF protein production and increased HIF-1 binding to the *MIF* promoter, while the activity of the *MIF* promoter was suppressed by mutations in the HRE and by deletion of the HRE in astrocytes.

**Funding:** This work was partly supported by grants from Japan Society of the Promotion of Science (JSPS) KAKENHI grant numbers 15K19727, 18K15483, and 21K07520 (SO), 15K09805 and 18K07556 (SB), as well as 17H04249 (AH), https://www.jsps.go.jp/j-grantsinaid/. The funders had no role in study design, data collection and analysis, decision to publish, or preparation of the manuscript.

**Competing interests:** The authors have declared that no competing interests exist.

## Conclusion

These results suggest that SNP rs17004038 in the HRE of the *MIF* promoter was significantly associated with SCZ and may be involved in the pathophysiology of SCZ via suppression of hypoxia and HIF pathway-induced MIF expression.

## Introduction

Schizophrenia (SCZ) is a chronic and disabling psychiatric disorder [1] that affects approximately 1% of the general population worldwide [2]. Although the onset of SCZ is commonly observed in adolescence, premorbid features, including cognitive impairments, often precede the diagnosis by years [3]. Recent genetic studies, including genome-wide association studies (GWASs), have contributed to the common theory that SCZ is a neurodevelopmental disorder [4–7]. According to the multiple-hit model, SCZ is attributable to the cumulative effects of genetic susceptibility and environmental insults during brain development [8]. Emerging evidence shows that gene–environment interactions underlie critical environmental contributions in the perinatal period [9–11]. In human research, there is consistent evidence linking perinatal hypoxia to SCZ in later life [12–16]. In addition, perinatal hypoxia has been reported to be a crucial environmental risk factor in the neurodevelopmental model of SCZ [17–20]. Indeed, oxygen restriction of perinatal murine has been established as an animal model of SCZ [21–23]. A recent review concluded that perinatal hypoxia, as well as the maternal–offspring immune activation and maternal hypothalamic–pituitary–adrenal axis, can contribute to the pathogenesis of SCZ based on early environmental upheavals, including prenatal maternal stress, obstetric complications, infections, and maternal lifestyle-related factors [24]. However, the detailed molecular mechanisms underlying the involvement of hypoxia in the pathophysiology of SCZ remain unclear.

Macrophage migration inhibitory factor (MIF) is a multifunctional cytokine that serves as a regulator in both innate and adaptive immunity. MIF is also expressed in nervous cells such as neurons [25, 26] and (predominantly) astrocytes [27–29]. In addition, MIF performs crucial functions in neurogenesis and neuroprotection [30], which are suggested to be involved in the pathophysiology of SCZ [31]. Several comprehensive biomarker studies have shown that MIF is a potential biomarker for SCZ [32, 33]. We also reported that the serum MIF level is higher in patients with SCZ and positively correlated with antipsychotic doses, and that a higher-expression allele of the $MIF$−794CATT$_{5-8}$ microsatellite (rs5844572) in the *MIF* promoter is significantly lower frequent in female patients with adolescent-onset SCZ [34]. Furthermore, we have shown that clozapine, an atypical antipsychotic, increased MIF expression via histone acetylation in neonatal mice-derived primary cultured astrocytes (PCAs) [35]. Taken together, these findings suggest that MIF may have an important role in the pathophysiology of SCZ and the functional mechanisms of antipsychotics.

The adaptive responses to hypoxia that restore oxygen homeostasis in tissues are predominantly regulated by hypoxia inducible factor (HIF)-1, a transcription factor activated by hypoxia [36, 37]. Recent studies have shown that hypoxia is a potent and rapid inducer of MIF expression via the HIF pathway, and that hypoxia-induced MIF expression is dependent on the hypoxia response element (HRE) in the 5′ UTR of the *MIF* gene [38, 39]. In addition, the single nucleotide polymorphism (SNP) rs17004038, which maps to the functional HRE in the *MIF* gene, prevents hypoxia-induced MIF expression [39].

Based on past studies including ours described above, we hypothesized that MIF may be involved in the pathophysiology of SCZ via the HIF pathway. To test this hypothesis, we first conducted an association study of SNP rs17004038 located in the HRE of the *MIF* promoter between patients with SCZ and controls. Subsequently, we investigated the molecular mechanisms underlying the effect of hypoxia on MIF expression in PCAs.

## Materials and methods

### Ethics statements

This study was conducted in accordance with the Declaration of Helsinki. The clinical part of this study was approved by the Ethical Committee for Genetic Studies of Kobe University Graduate School of Medicine and the Ethics Committee of Genetics at Niigata University. Written informed consent from all participants was obtained after the details of the procedures of the clinical part of this study had been fully explained. The animal study protocol was approved by the Institutional Animal Care and Use Committee and carried out according to the Kobe University Animal Experimentation Regulations. All efforts were made to minimize suffering of mice.

### Participants

In the association study, all participants were of Japanese descent and recruited from the suburbs of Kobe city (first set, 915 patients and 836 controls) and Niigata city (second set, 843 patients and 671 controls) in Japan. A diagnosis of SCZ was made by at least two psychiatrists according to the DSM-5 criteria for SCZ. Control participants were healthy volunteers who were screened for psychiatric disorders by a psychiatrist. None of the control participants had any present, past, or family (first-degree relatives) history of psychiatric disorders, use of neuroleptic medication, and substance abuse outside of nicotine. The demographic and clinical characteristics are shown in S1 Table.

### Genotyping of SNP rs17004038 in the HRE of the *MIF* gene

Genotyping was performed as described elsewhere [34, 40]. The *MIF* gene is located on chromosome 22q11.23 (GenBank accession No. NM_002415). We analyzed SNP rs17004038 (C>A), which is located in the HRE of the *MIF* promoter and is involved in hypoxia-induced MIF expression [39] (Fig 1).

Peripheral blood samples were obtained from the participants, and DNA was extracted with the QIAamp DNA Blood Midi Kit (Qiagen Inc., Valencia, CA, USA). The quantity and purity of the DNA were determined with a NanoDrop spectrophotometer (Thermo Fisher Scientific, Waltham, MA, USA). DNA samples were kept frozen at −80˚C until analysis. We obtained the pre-designed TaqMan SNP genotyping assay for SNP rs17004038 from the Applied Biosystems database and performed genotyping with a 7500 Real-Time PCR System (Applied Biosystems) in accordance with the manufacturer's protocol.

### Cell culture

PCAs were prepared from neonatal C57BL/6J mice as described elsewhere [35, 41]. Briefly, the isolated forebrain was minced and incubated with DNase I (FUJIFILM Wako Pure Chemical) and trypsin (Thermo Fisher Scientific) for dissociation. The dissociated tissues were suspended in Dulbecco's modified Eagle medium/Nutrient Mixture F-12 (DMEM/F12; Thermo Fisher Scientific) supplemented with 10% fetal bovine serum (FBS; Sigma-Aldrich, St. Louis, MO, USA) and 1% penicillin/streptomycin (PS; Thermo Fisher). The suspended cells were seeded

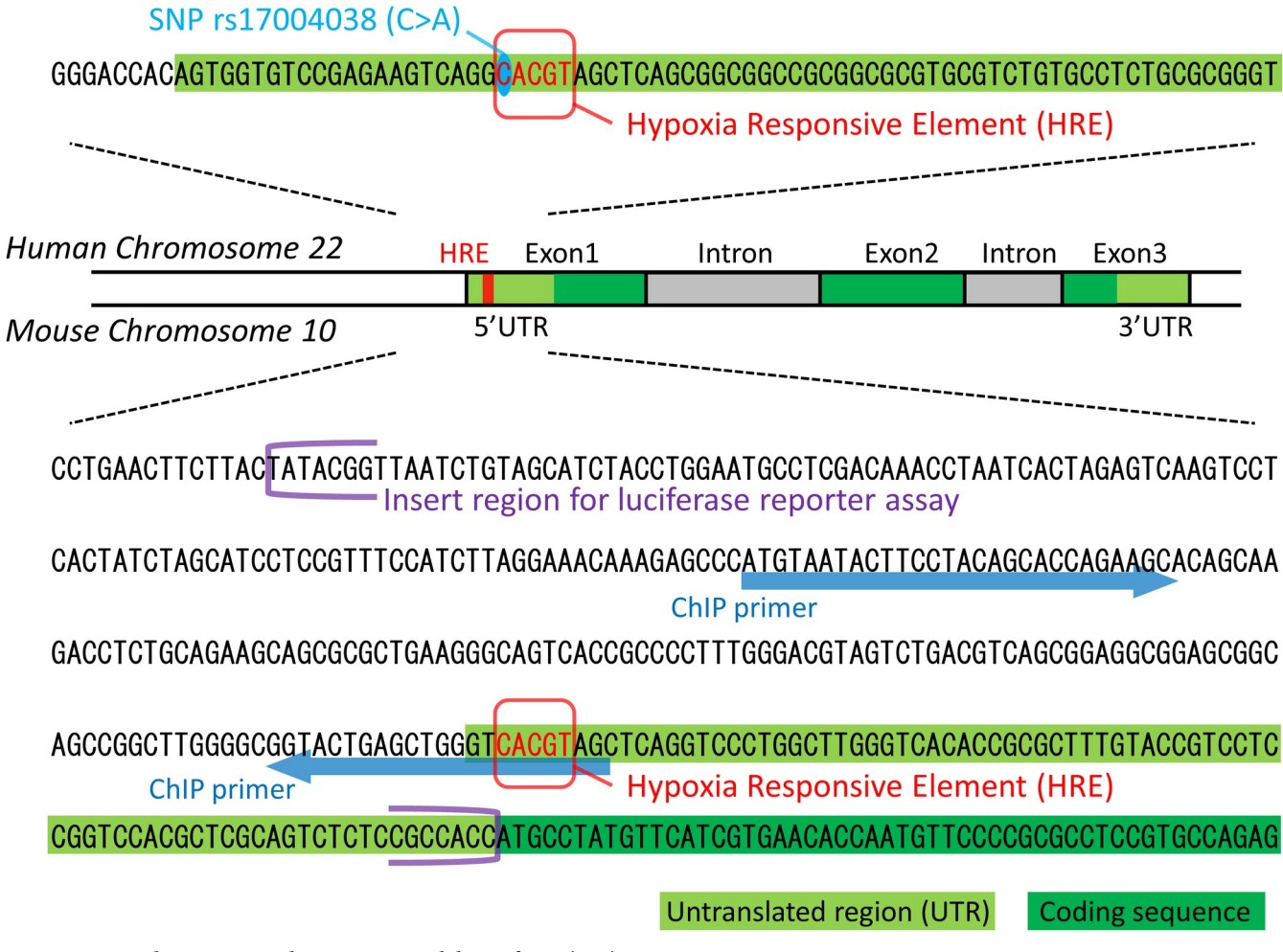

**Fig 1. Human and mouse macrophage migration inhibitory factor (*MIF*) gene promoter region.**

in poly-L-lysine (PLL; ScienCell, Carlsbad, CA, USA)-coated 75-mm$^2$ flasks ($10–15 \times 10^6$ cells/flask). The plated cells were cultured in a 5% $CO_2$ incubator at 37˚C. Every 8–12 days, the confluent cells were shaken for 10 min to separate them from microglial cells, released from the flasks with trypsin, and seeded onto non-coated flasks. These cells were seeded onto non-coated 8-well chamber slides ($2.5 \times 10^4$ cells/well) and evaluated with immunocytochemistry for anti-glial fibrillary acidic protein (GFAP) antibody (Abcam).

## Drugs

We used ML228 (Tocris Bioscience, Bristol, UK) as a HIF pathway activator and YC-1 (Abcam, Cambridge, UK) as a HIF inhibitor.

## Hypoxic treatments

For hypoxic treatments, we applied the BIONIX-1 hypoxic culture kit (Sugiyamagen, Tokyo, Japan) as previously described [42, 43]. This system consists of an AnaeroPack-Anaero 5%

(oxygen absorber; Mitsubishi Gas Chemical, Tokyo, Japan), an OXY-1 oxygen monitor (JIKCO, Tokyo, Japan), an AnaeroPouch (Mitsubishi Gas Chemical), and plastic clips for sealing the pouch. Briefly, PCAs, the oxygen absorber, and the oxygen monitor were arranged in the pouch, and the left open side was sealed with a clip. The $O_2$ concentration in the pouch rapidly decreased after sealing; once a low $O_2$ concentration (approximately 0.1%) was reached, the pouch was sealed with another clip between the culture dish and the oxygen absorber to stop further oxygen absorption. Then, the pouch was maintained in an incubator at 37˚C.

## Total RNA extraction and quantitative reverse transcription polymerase chain reaction (qRT-PCR) assay

PCAs were seeded onto non-coated 6-well plates ($2 \times 10^5$ cells/well) in DMEM/F12 supplemented with 10% FBS and PS. After 3 days, the medium was changed to FBS-free medium. After overnight incubation, drug administration or hypoxic treatments were performed and followed by incubation. Total RNA was extracted from PCAs with RNeasy Mini Kit (Qiagen, Hilden, Germany), followed by conversion to cDNA with Quantitect Reverse Transcription Kit (Qiagen). Quantitative PCR was performed with TB Green Advantage qPCR Premix (Takara Bio, Tokyo, Japan) and the QuantiStudio 3 Real-Time PCR System (Applied Biosystems, Foster City, CA, USA) according to the manufacturer's protocol. Beta-actin (ACTB) was used as an endogenous control. The sequences of the forward and reverse primers are described in S2 Table. The PCR conditions were as follows: 95˚C for 30 s, followed by 40 cycles of 95˚C for 5 s and 60˚C for 34 s.

## Enzyme-linked immunosorbent assay (ELISA)

ELISA was performed as described elsewhere [35]. Briefly, PCAs were seeded onto non-coated 6-well plates ($2 \times 10^5$ cells/well) in DMEM/F12 supplemented with 10% FBS and PS. After 3 days, the culture medium was changed to an FBS-free medium. After overnight incubation, hypoxic treatment was performed and followed by incubation. The culture medium and cell lysate were collected from PCAs and stored at −80˚C. Total protein concentration of the samples was quantified using Qubit 3.0 Fluorometer (Thermo Fisher Scientific). HIF-1α protein concentration was measured with the Mouse HIF-1-alpha SimpleStep ELISA Kit (Abcam, Cambridge, UK) according to the manufacturer's protocol. MIF protein concentration was measured with the Mouse MIF SimpleStep ELISA Kit (Abcam) according to the manufacturer's protocol. The absorbance at 450 nm was detected with Multiskan FC (Thermo Fisher Scientific). All samples and standards were measured in duplicate. The protein level per 1 μg of total protein was calculated.

## Chromatin immunoprecipitation (ChIP) assay

The ChIP assay was performed as described elsewhere [35]. Briefly, PCAs were seeded onto a non-coated 90-mm dish ($2 \times 10^6$ cells/dish) in DMEM/F12 supplemented with 10% FBS and PS. After 3 days, the culture medium was changed to FBS-free medium. After overnight incubation, hypoxic treatment was performed and followed by incubation. For the ChIP assay, we used the ChIP-IT express kit (Active Motif, La Hulpe, Belgium) according to the manufacturer's protocol. Briefly, PCAs were fixed with 37% formaldehyde. For shearing the chromatin of the fixed PCAs, sonication was performed with Bioruptor II (BM Equipment, Tokyo, Japan) over 10 homogenization cycles. Each homogenization cycle consisted of homogenization at the maximum setting for 30 s followed by a 30 s interval. The sheared chromatin was utilized for immunoprecipitation with the anti-HIF-1α antibody (ab1, Abcam). Real-time PCR for the

immunoprecipitated DNA was performed with TB Green Advantage qPCR Premix (Takara Bio) and the QuantiStudio 3 Real-Time PCR System (Applied Biosystems). The forward and reverse primers were designed to cover 151 base pairs of the *MIF* promoter (Fig 1 and S2 Table). The PCR conditions were as follows: 95˚C for 30 s, followed by 40 cycles of 95˚C for 5 s, 60˚C for 20 s and 72˚C for 34 s.

## Plasmid construction

We subcloned a part of the *MIF* promoter region (−335 to −1), which included the HRE, as the insert region for the luciferase reporter assay (Fig 1). The forward and reverse primers ware designed with the online Primer Design tool for In-Fusion (S2 Table). Mouse genomic DNA extracted from PCAs with the AllPrep DNA/RNA Mini Kit (Qiagen) was used as the template for PCR amplification of the insert region. The reporter plasmid was generated by incorporating the PCR-amplified insert region upstream of the Nanoluc luciferase reporter gene of the pNL1.2 [*NlucP*] vector (Promega, Madison, WI, USA) with the In-Fusion HD Cloning Kit (Takara Bio) and *Escherichia coli* Competent High DH5α (Toyobo, Osaka, Japan), according to the manufacturer's protocol. All of the mutated HRE plasmids, such as the SNP HRE, the mutant HRE, and the deletion HRE, were generated by site-directed mutagenesis with the KOD Plus Mutagenesis Kit (Toyobo). The primers harboring the desired mutations are shown in S2 Table.

## Luciferase reporter assay

PCAs were seeded onto non-coated 6-well plates ($2 \times 10^5$ cells/well) in DMEM/F12 supplemented with 10% FBS and PS. After 3 days, the medium was changed to FBS-free medium. After overnight incubation, each well was transfected with a DNA mixture containing 1.25 μg of the indicated reporter plasmid and 1.25 μg of pGL4.13[*luc2*/SV40] Vector (Promega) encoding the firefly luciferase under the control of a SV40 promoter using Lipofectamine LTX Reagent with PLUS Reagent (Thermo Fisher Scientific). At 24 h after transfection, PCAs were transferred to hypoxic conditions (0.1% $O_2$) or left under normoxic conditions for 24 h. Cell lysate was collected from PCAs with Passive Lysis Buffer (Promega) and stored at −80˚C. NanoLuc and firefly luciferase activities were determined with Nano-Glo Dual-Luciferase Reporter Assay (Promega) and GloMax Navigator Microplate Luminometer (Promega) according to the manufacturer's protocol. In order to correct for transfection efficiency, the NanoLuc luciferase activity was normalized to the firefly luciferase activity.

## Statistical analyses

Statistical analyses were performed with R version 3.6.1 (The R Foundation for Statistical Computing, Vienna, Austria) and EZR version 1.41 (Jichi Medical University, Saitama, Japan) [44]. For the SNP rs17004038 association study, we used Haploview version 4.2 (Daly Lab at the Broad Institute Cambridge, MA, USA) [45] to determine the Hardy–Weinberg equilibrium (HWE), allele frequencies, and genetic associations. Genotypic association was examined with the Cochran-Armitage trend test. Allelic association was examined with the $\chi^2$ test. For the PCA study, the group differences were analyzed using one-way ANOVA with Dunnett's or Tukey's multiple comparison tests, as appropriate. Statistical significance was defined as two-tailed *p*-value < 0.05.

# Results

## Association study of SNP rs17004038 in the HRE of the human *MIF* gene

The distribution of SNP rs17004038 (C>A) in patients with schizophrenia and controls was investigated. We found a non-significant trend for the association of SNP rs17004038 with

**Table 1. Distribution of rs17004038 in patients with schizophrenia and controls in this study.**

| | n | HWE | Genotype | | | | Allele | | | | | |
| | | | C/C | C/A | A/A | P-value[a] | C | A | MAF | P-value[b] | Odds ratio (95% CI) | Power |
|---|---|---|---|---|---|---|---|---|---|---|---|---|
| Overall | | | | | | | | | | | | |
| SCZ | 1758 | 1.00 | 1677 | 80 | 1 | **0.0467** | 3434 | 82 | 0.0233 | **0.0424** | 1.445 (1.011–2.065) | 0.520 |
| CTL | 1507 | 0.012 | 1461 | 43 | 3 | | 2965 | 49 | 0.0163 | | | |
| Male | | | | | | | | | | | | |
| SCZ | 946 | 0.77 | 904 | 41 | 1 | 0.389 | 1849 | 43 | 0.0227 | 0.368 | 1.250 (0.769–2.032) | 0.141 |
| CTL | 739 | 0.0025 | 715 | 21 | 3 | | 1451 | 27 | 0.0183 | | | |
| Female | | | | | | | | | | | | |
| SCZ | 812 | 1.00 | 773 | 39 | 0 | **0.0456** | 1585 | 39 | 0.0240 | **0.0478** | 1.693 (0.999–2.869) | 0.509 |
| CTL | 768 | 1.00 | 746 | 22 | 0 | | 1514 | 22 | 0.0143 | | | |

CI, Confidence interval; CTL, control; HWE, Hardy-Weinberg equilibrium; MAF, minor allele frequency; SCZ, schizophrenia.

Boldface type indicates statistical significance.

[a] Genotypic p-values were calculated with Cochran-Armitage trend test.

[b] Allelic p-values were calculated with the $\chi^2$ test.

SCZ in the first set of subjects ($p = 0.0509$) (S3 Table), but not in the second set ($p = 0.364$) (S4 Table). Next, we performed an analysis using both sets combined and found a significant association of SNP rs17004038 with SCZ ($p = 0.0424$). The odds ratio (1.445) suggests that the minor A allele of SNP rs17004038 may be involved in the pathophysiology of SCZ (Table 1).

We additionally performed subgroup analyses divided by sex. Each set of subjects showed no significant difference in the frequency of the allele between SCZ and control (S3 and S4 Tables). Combining the sets revealed a significant difference in the samples from female subjects ($p = 0.0478$) but not male subjects ($p = 0.368$), consistent with the overall analysis (Table 1).

## Effects of hypoxia on mouse *HIF* mRNA expression and HIF protein level

First, the effect of hypoxia on *HIF-1α/β* mRNA expression was examined. The *HIF-1α/β* mRNA expression levels were slightly changed by hypoxia, although significant changes were observed in *HIF-1α* mRNA level at 48 h and *HIF-1β* mRNA level at 24 h (S1A and S1B Fig). Next, the effect of hypoxia on HIF-1α protein level was examined with ELISA. Hypoxic treatments for 24 and 48 h increased HIF-1α protein level in the cell lysates (S1C Fig).

## Effects of hypoxia on mouse *MIF* mRNA expression

First, the effect of ML228, a HIF pathway activator, on *MIF* mRNA expression was examined. ML228 increased *MIF* mRNA expression in a dose-dependent manner 48 h after its administration (Fig 2A). Next, the effect of hypoxia on *MIF* mRNA expression was examined. Hypoxia increased *MIF* mRNA expression in a time-dependent manner, which reached significance at 6 h and increased until 48 h after the start of hypoxic treatment (Fig 2B). On the other hand, YC-1, a HIF inhibitor, significantly suppressed *MIF* mRNA expression 6 h after the start of hypoxic treatments (Fig 2C).

## Effects of hypoxia on mouse MIF protein production

To examine whether hypoxia increases MIF protein production as in the case of mRNA, the effect of hypoxia on MIF protein production was examined with ELISA. Hypoxic treatments

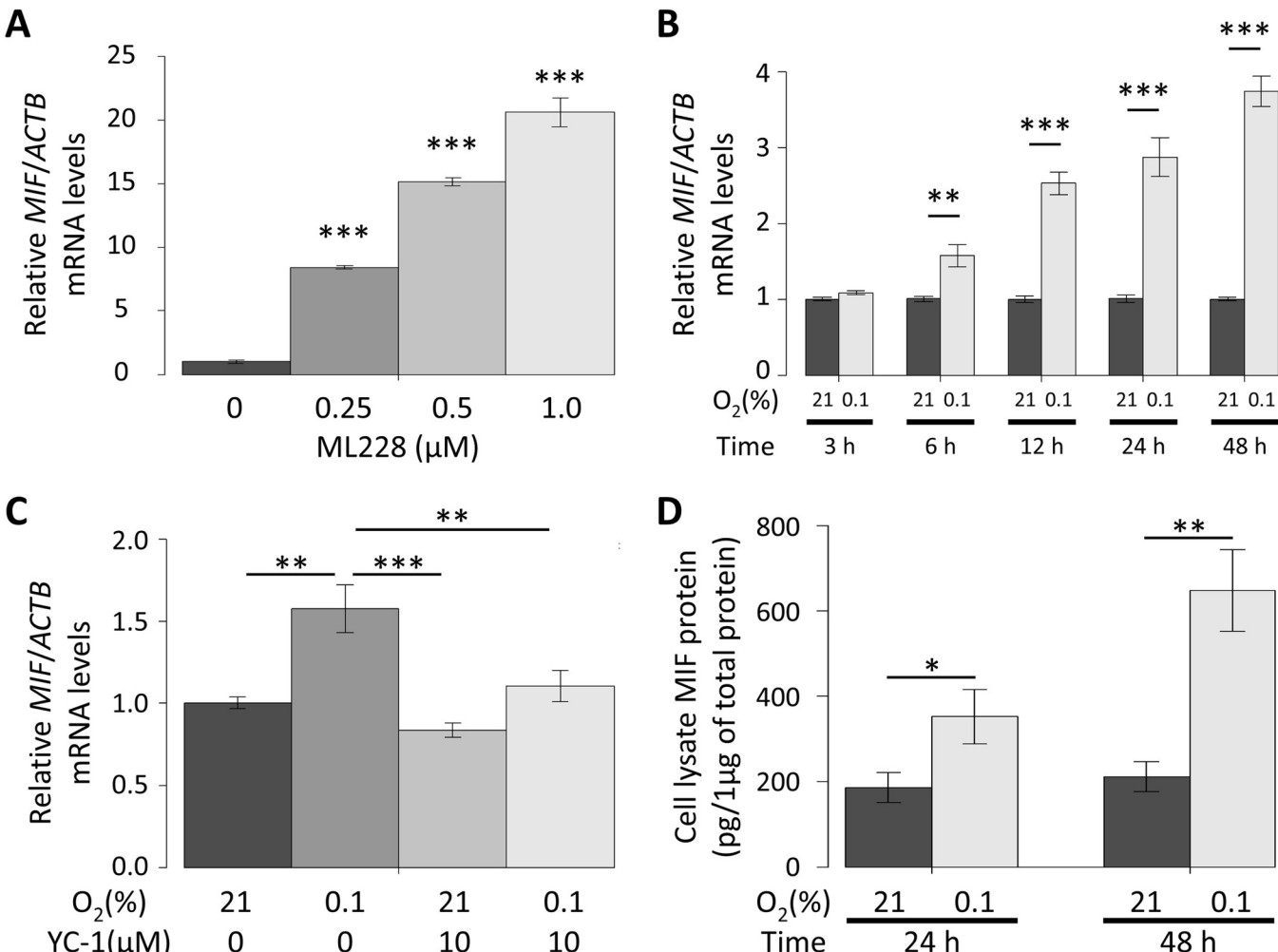

**Fig 2. Hypoxia induced macrophage migration inhibitory factor (*MIF*) mRNA expression and protein production via the hypoxia inducible factor (HIF) pathway. A.** The effects of ML228, a HIF activator, on *MIF* mRNA expression in primary cultured astrocytes (PCAs). PCAs were incubated for 48 h in 0, 0.25, 0.5, or 1.0 μM ML228. *MIF* mRNA expression was analyzed by qRT-PCR. The values are shown as the ratio of *MIF* mRNA to beta-actin (*ACTB*) mRNA (Dunnett's test vs. vehicle; n = 3). **B.** The time-dependent effects of hypoxia on *MIF* mRNA expression in PCAs. PCAs were incubated for 3, 6, 12, 24, or 48 h in 21.0% or 0.1% $O_2$. *MIF* mRNA expression was analyzed by qRT-PCR. The values are shown as the ratio of *MIF* mRNA to *ACTB* mRNA (Student's t-test; n = 5–6). **C.** The effects of the HIF inhibitor YC-1 on hypoxia-induced *MIF* mRNA expression in PCAs. PCAs were incubated for 6 h in 21.0% or 0.1% $O_2$ and treated with 0 or 10 μM of YC-1. *MIF* mRNA expression was analyzed by qRT-PCR. The values are shown as the ratio of *MIF* mRNA to *ACTB* mRNA (Tukey's test; n = 6). **D.** The effects of hypoxia on MIF protein production in cell lysate of PCAs. PCAs were incubated for 24 or 48 h in 21.0% or 0.1% $O_2$. The MIF protein level per 1 μg of total protein was analyzed using a MIF enzyme-linked immunosorbent assay (ELISA) (Student's t-test; n = 6). The data are expressed as the mean ± SEM. $^*p < 0.05$, $^{**}p < 0.01$, or $^{***}p < 0.001$.

for 24 and 48 h increased MIF protein production in the cell lysates (Fig 2D) but not in the cell culture medium (S2 Fig).

## Effects of hypoxia on HIF-1α binding to the HRE of mouse *MIF* promoter

Under hypoxic conditions, the HIF-1α subunit translocates to the cell nucleus and associates with the HIF-1β subunit, and then this HIF-1α/1β complex binds to the HRE of the promoter region of its target genes. Therefore, we investigated the effects of hypoxia on the binding of HIF-1α to the *MIF* promoter with the ChIP assay. We found a significant increase in HIF-1α binding to the *MIF* promoter 6 h after the start of hypoxic treatments but not after 24 or 48 h (Fig 3A).

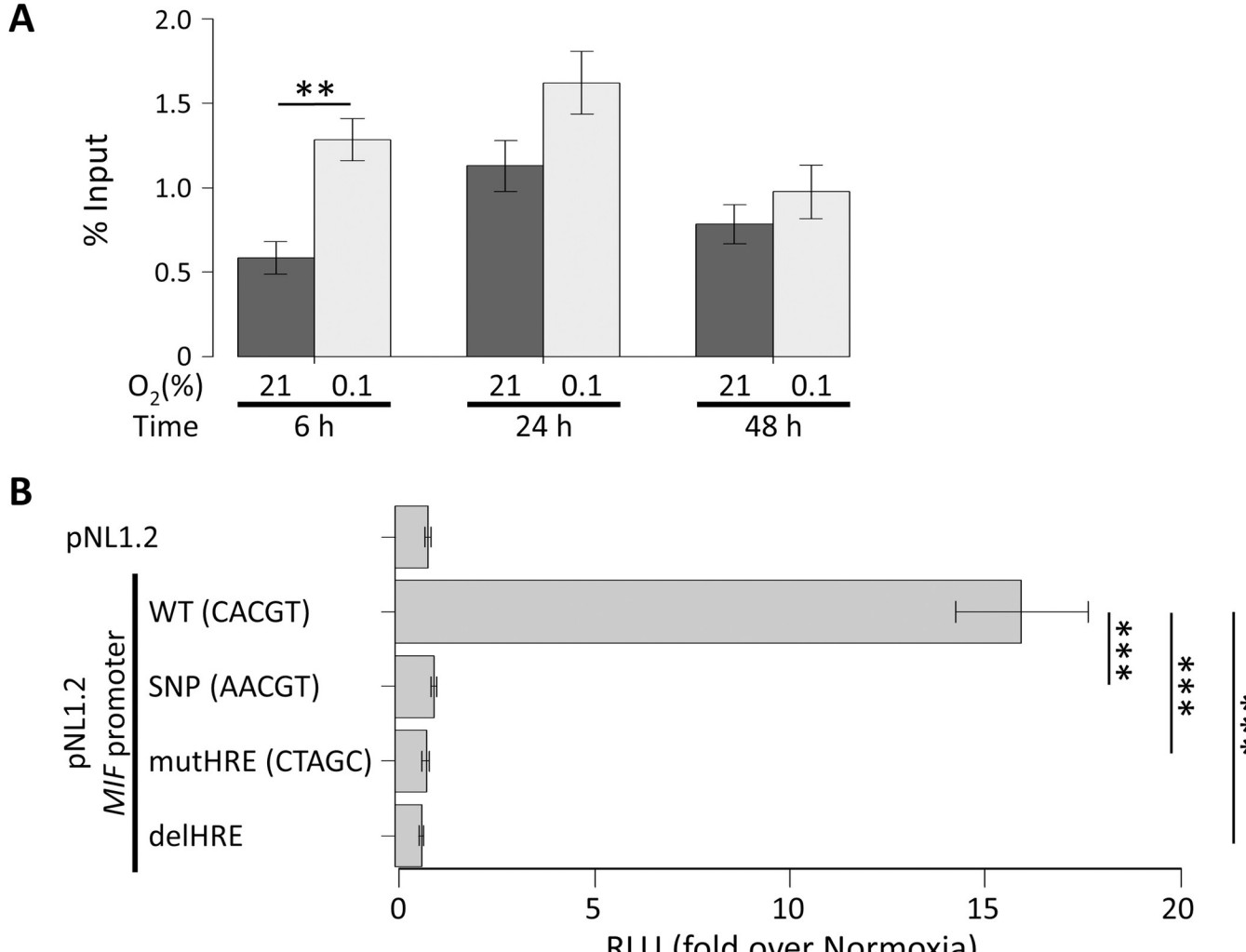

**Fig 3. Hypoxia-inducible factor (HIF) binding to the hypoxia response element (HRE) induced activation of the macrophage migration inhibitory factor (*MIF*) gene promoter under hypoxia, that was abrogated by the allelic variant of the HRE. A.** The effects of hypoxia on HIF binding at the *MIF* gene promoter in primary cultured astrocytes (PCAs). PCAs were incubated for 6, 24, or 48 h in 21.0% or 0.1% $O_2$. HIF-1α binding to the *MIF* gene promoter was analyzed by chromatin immunoprecipitation (ChIP) assay. Real-time PCR was performed on DNA purified from each of the ChIP reactions by using a primer set specific for the *MIF* gene promoter. The % input values indicate the ratio of immunoprecipitated DNA fragments to the input DNA fragments (Student's t-test after logarithmic transformation; n = 6). **B.** The allelic variant of the HRE abrogates MIF induction by hypoxia in PCAs. PCAs were transfected with a reporter plasmid containing the *MIF* genomic region (−335 to −1) upstream of the Nanoluc luciferase reporter gene and a control plasmid containing the firefly luciferase reporter gene. Where indicated, the wild type (WT) HRE sequence (CACGT) was mutated to AACGT (SNP), CTAGC (mutHRE), or deletion (delHRE). At 24 h after transfection, PCAs were incubated for 24 h in 21.0% or 0.1% $O_2$. The graph represents the corrected luciferase activity values of each construct in cells exposed to hypoxia over the luciferase activity obtained in normoxic cells (Dunnett's test; n = 9–14). The data are expressed as the mean ± SEM. *$p < 0.05$, **$p < 0.01$, or ***$p < 0.001$.

## Effects of the SCZ-associated SNP in the HRE on hypoxia-induced activation of the *MIF* promoter

The adaptation to hypoxia is mainly dependent on HIF-mediated gene expression, which means that the effect of the SCZ-associated SNP on the HRE of the *MIF* promoter might alter the response to hypoxia in individuals with this SNP. The HRE sequence (CACGT) of the *MIF* promoter identified in the human genome also exists in the mouse genome (Fig 1). In order to investigate the biological effect of the SCZ-associated SNP rs17004038, we constructed

plasmids for a luciferase reporter assay including the wild-type mouse *MIF* promoter, the C→A variant corresponding to the SNP rs17004038, and mutation/deletion of the HRE placed upstream of the Nanoluc luciferase gene. Then, we performed luciferase reporter assays with these plasmids. Like previous studies using human cell lines [38, 39], the activity of mouse *MIF* promoter (WT) was robustly promoted by hypoxia in PCAs (Fig 3B). In contrast, mutation (mutHRE) and deletion (delHRE) versions of the HRE completely suppressed hypoxia-induced activation of *MIF* promoter (Fig 3B). In addition, the variant allele C→A also completely suppressed hypoxia-induced activation of *MIF* promoter as effectively as mutHRE and delHRE (Fig 3B).

## Discussion

First, we investigated the SNP rs17004038 (C>A) in the HRE of the *MIF* promoter. We found that rs17004038 is significantly associated with SCZ. The odds ratio indicates the minor A allele of rs17004038 is a risk allele for SCZ. Sex distribution was significantly unequal between control and SCZ groups in the first set of participants. Thus, we additionally performed sub-group analyses divided by sex. The samples from female subjects showed a significant difference, consistent with overall results, but the samples from male subjects did not. One reason is that the sample size of male subjects was relatively small, considering that the same minor allele frequency was present in both samples of male (SCZ: 0.0227 vs. CTL: 0.0183) and female (SCZ: 0.0240 vs. CTL: 0.0143) subjects, and that the samples from male subjects had smaller sample power (0.141) than that of female subjects (0.509). Another reason may be the influence of sex difference in the development of SCZ. Further studies with larger samples are required.

Next, we investigated the functional role of the SNP rs17004038 on the HRE of the *MIF* promoter by performing biological experiments with neonatal mice-derived PCAs. First, we confirmed that hypoxia increased HIF-1α protein level in the cell lysates of PCAs, although *HIF-1α/β* mRNA levels were little changed. Under normoxic conditions, the intracellular concentration of HIF-1α protein is low, not due to lower protein expression but because it is negatively regulated by proteolysis by the ubiquitin-proteasome system via the von Hippel–Landau tumor suppressor; under hypoxic conditions, HIF-1α levels are stable and it translocates to the cell nucleus and associates with HIF-1β, and the HIF-1α/1β complex binds to the HRE of the DNA resulting in the transcription of various genes [46–48]. Our experimental system using PCAs under hypoxia seems to be in agreement with the results of previous studies.

Both hypoxia and ML228, a HIF activator, increased *MIF* mRNA expression. Hypoxia also induced MIF protein production in cell lysates. Conversely, YC-1, a HIF inhibitor, disturbed hypoxia-induced *MIF* mRNA expression. In addition, a ChIP assay showed that hypoxia increased the binding of the HIF-1α subunit to the *MIF* promoter region including the HRE. These findings strongly indicate that the HRE plays a role in the regulation of MIF expression. Finally, the luciferase reporter assay showed that hypoxia-induced activation of the *MIF* promoter was disrupted by variants of the HRE including that corresponding to the SNP rs17004038, which suggests that SNP rs17004038 may be involved in the pathophysiology of SCZ through disruption of hypoxia-induced MIF expression. Our present study is the first study to show the involvement of MIF in hypoxia-associated mechanisms of SCZ.

Previous studies have shown that blood MIF is increased in patients with SCZ, including initial-onset, drug-naive patients [32, 33]. Our previous studies showed that antipsychotic dosage positively correlated with MIF blood levels, that a higher-expression polymorphism of the *MIF* promoter is significantly lower frequent in female patients with adolescent-onset SCZ [34], and that clozapine and other antipsychotics increased MIF expression in PCAs [35].

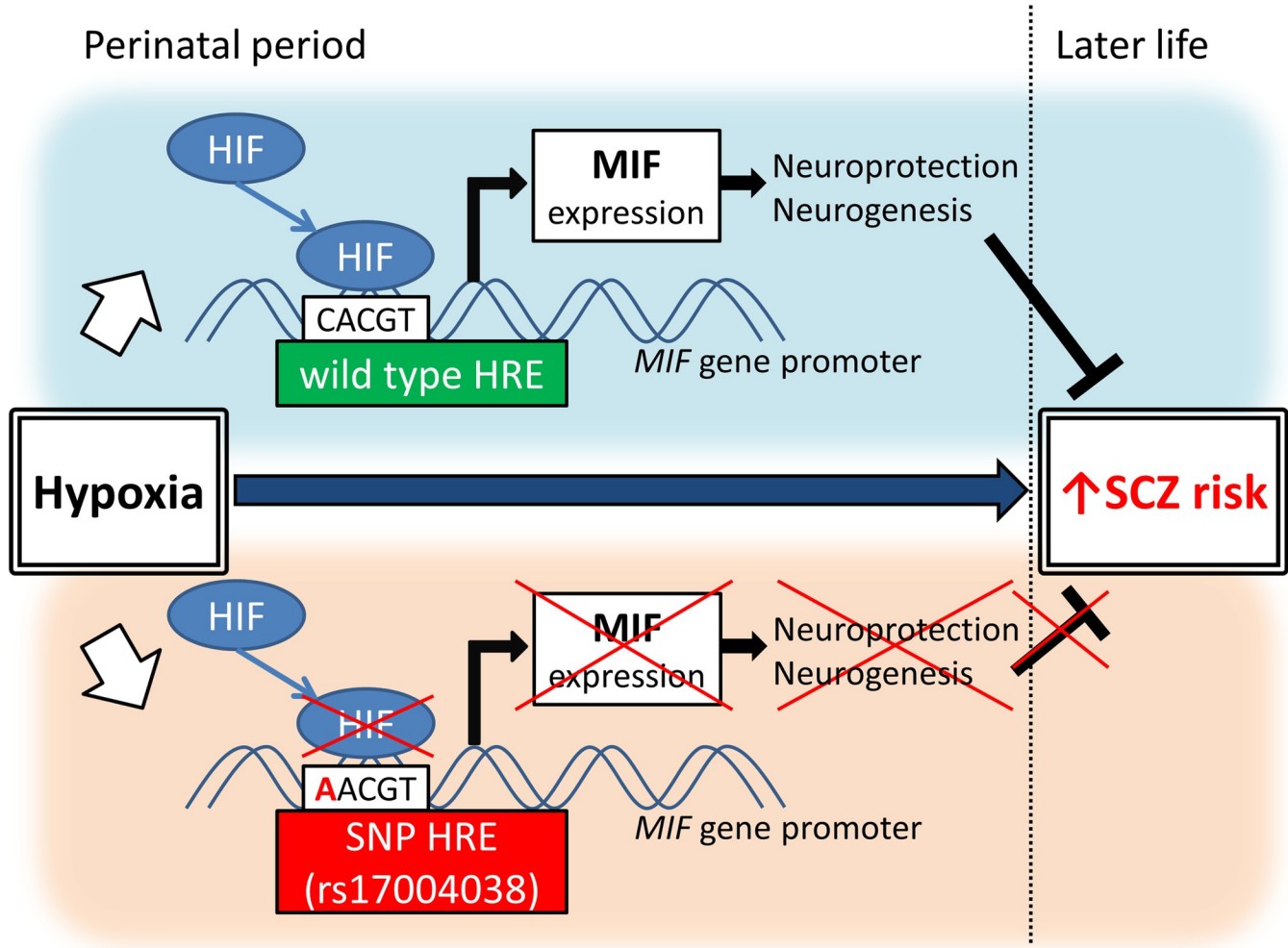

**Fig 4. Putative roles of single nucleotide polymorphism (SNP) rs17004038 in the hypoxia response element (HRE) of the macrophage migration inhibitory factor (*MIF*) gene promoter in perinatal hypoxia-associated schizophrenia (SCZ) risk.** Abbreviation: HIF, Hypoxia-inducible factor.

These findings suggest that MIF may function as a physiological protective factor against SCZ risk [35]. The present study showed that SNP rs17004038 in the HRE of the *MIF* gene is significantly more common in patients with SCZ, and that this SNP inhibits hypoxia-induced MIF expression in PCAs from mice. Taken together, these studies suggest that elevated levels of hypoxia-induced MIF may have a protective function against perinatal hypoxia-associated SCZ development, and that SNP rs17004038 may inhibit the protective function of MIF by blocking MIF expression in response to hypoxia. MIF may play a role in the pathophysiology of SCZ through both genetic factors (i.e., the SNP) and environmental factors (i.e., hypoxia), indicating that MIF may mediate the gene-environment interaction underlying the pathophysiology of SCZ. Putative roles of SNP rs17004038 in the HRE of the *MIF* gene promoter in perinatal hypoxia-associated SCZ risk are shown in Fig 4.

MIF is widely expressed during development in the nervous system [49, 50], which suggests that it plays a role in neural development. In fact, MIF is predominantly expressed in astrocytes, a major regulator of hippocampal neurogenesis [51], in the brain; regulates the proliferation, differentiation, and survival of neural stem/progenitor cells [30, 52–54]; and plays a role

in hippocampal development [55]. Interestingly, the hippocampus is one of the most hypoxia-sensitive regions in the brain [56] and a major SCZ-associated brain region [31]. In addition, perinatal hypoxia is related to the risk of SCZ [12–16]. Taken together, these studies easily lead to the hypothesis that perinatal hypoxia increases the risk of SCZ by affecting MIF-mediated hippocampal development. Our present results showing that a SCZ-associated SNP in the HRE of the *MIF* promoter disrupted hypoxia-induced MIF expression in neonatal mice-derived astrocytes can be considered to validate this hypothesis.

Although we showed the association of SNP rs17004038 (C>A) with SCZ, recent large GWASs of SCZ did not identify this SNP [5–7]. We showed that the A allele frequency of rs17004038 is around 2%. This result is consistent with the Japanese data from the 1000 Genomes Project database (https://www.ncbi.nlm.nih.gov/variation/tools/1000genomes/) [57]. However, the A allele is found only in Asian populations, and the global A allele frequency is 0.0044 (S5 Table). Thus, rs17004038 may not have been adopted in SNP arrays used in previous GWASs because its minor allele frequency is relatively low. Therefore, our findings for SNP rs17004038 indicate the need for further studies that also consider such relatively rare variants.

Hypoxia is a common phenomenon in malignant tumors, and its roles in cancer progression, angiogenesis, and metastasis have been well studied [58]. In addition, numerous studies have shown that MIF plays an important role in advocating such tumorigenic processes [59, 60], and that HIF-induced MIF expression is associated with tumorigenesis [38]. On the other hand, hypoxia may be involved in neurological diseases such as ischemic stroke, and a few studies reported that MIF plays a beneficial role in neurological recovery after ischemic stroke [61–63]. Moreover, recombinant MIF administration is effective for inducing the expression of brain-derived neuro-trophic factor and promotes neuroprotection of neuronal cells in human neuroblastoma cell lines under oxygen-glucose deprivation [64]. A recent review concluded that the role of MIF in neurological disorders is controversial and depends on the pathophysiological context and/or cellular microenvironment [65]. Considering our results and the findings of previous studies, further elucidation is required for not only the pathological but also neuroprotective role of MIF in neurological and psychiatric diseases via HIF pathway under hypoxic conditions.

Since hypoxia increased *MIF* mRNA expression and MIF is a secreted protein, we predicted that hypoxia may increase the MIF protein level in cell culture medium. However, contrary to our expectation, hypoxia increased the MIF protein level in the cell lysate of PCAs but not in the cell culture medium. MIF is a type of cytokine, and produced cytokines are usually stored in secretory vesicles or granules via the Golgi and then secreted by exocytosis in response to various stimuli [66]. Merk et al. showed that p115, a Golgi-associated protein, binds to MIF and facilitates its secretion [67]. These findings suggest that hypoxia may directly increase MIF protein production in astrocytes but does not affect its secretion, and that some indirect path-ways may be involved in MIF protein secretion in actual brain tissues.

Our study had several limitations. First, our association study included only a Japanese population. Therefore, the results of our association study might not be generalizable to other populations. Second, we used mice-derived cells. Thus, the findings should be validated in analyses focusing on human astrocytes performed using cell lines derived from human brain tissues or induced-pluripotent stem cells. Third, the present study is an *in vitro* study. Therefore, the role of MIF and the HIF pathway in the pathophysiology of SCZ under hypoxia should be further investigated in *in vivo* models of SCZ.

## Conclusions

We have shown that a SNP of the HRE in the *MIF* promoter is associated with SCZ, and that this SNP disrupts hypoxia-induced MIF expression via the HIF pathway in astrocytes. These

findings indicate the potential role of astrocyte-derived MIF in the hypoxia-related pathophysiology of SCZ.

## Supporting information

**S1 Fig. Hypoxia induced hypoxia inducible factor (*HIF*)-*1α/β* mRNA expression and protein level. A.** The time-dependent effects of hypoxia on *HIF-1α* mRNA expression in primary cultured astrocytes (PCAs). PCAs were incubated for 3, 6, 12, 24, or 48 h in 21.0% or 0.1% $O_2$. *HIF-1α* mRNA expression was analyzed by qRT-PCR. The values are shown as the ratio of *HIF-1α* mRNA to beta-actin (*ACTB*) mRNA (Student's t-test; n = 6). **B.** The time-dependent effects of hypoxia on *HIF-1β* mRNA expression in PCAs. PCAs were incubated for 3, 6, 12, 24, or 48 h in 21.0% or 0.1% $O_2$. *HIF-1β* mRNA expression was analyzed by qRT-PCR. The values are shown as the ratio of *HIF-1β* mRNA to *ACTB* mRNA (Student's t-test; n = 6). **C.** The effects of hypoxia on HIF-1α protein level in cell lysate of PCAs. PCAs were incubated for 24 or 48 h in 21.0% or 0.1% $O_2$. The HIF-1α protein level per 1 μg of total protein was analyzed using a HIF-1α enzyme-linked immunosorbent assay (ELISA) (Student's t-test; n = 6). The data are expressed as the mean ± SEM. $^*p < 0.05$, $^{**}p < 0.01$, or $^{***}p < 0.001$.
(TIF)

**S2 Fig. The effects of hypoxia on macrophage migration inhibitory factor (MIF) protein production in cell culture medium of primary cultured astrocytes (PCAs).** PCAs were incubated for 24 or 48 h in 21.0% or 0.1% $O_2$. The MIF protein level per 1 μg of total protein was analyzed using a MIF enzyme-linked immunosorbent assay (ELISA). The data are expressed as the mean ± SEM. $^*p < 0.05$, $^{**}p < 0.01$, or $^{***}p < 0.001$ (Student's t-test; n = 6).
(TIF)

**S1 Table. Demographic and clinical characteristics of participants for the macrophage migration inhibitory factor gene polymorphism association study.**
(DOCX)

**S2 Table. Primers used in the experiments with primary cultured astrocytes from mouse.**
(DOCX)

**S3 Table. Distribution of rs17004038 in patients with schizophrenia and controls in the first set of subjects.**
(DOCX)

**S4 Table. Distribution of rs17004038 in patients with schizophrenia and controls in the second set of subjects.**
(DOCX)

**S5 Table. Allele frequency of SNP rs17004038 in different ethnic populations.**
(DOCX)

**S1 Data.**
(XLSX)

## Acknowledgments

We thank Y. Nagashima and H. Maeda for their expert technical assistance, as well as M. Takebayashi, K. Hisaoka-Nakashima, and N. Kajitani for their helpful advice about primary cultured astrocytes.

## Author Contributions

**Conceptualization:** Satoshi Okazaki, Shuken Boku, Atsushi Kimura.

**Data curation:** Satoshi Okazaki, Shuken Boku, Yuichiro Watanabe, Ikuo Otsuka, Tadasu Horai, Ryo Morikawa, Naofumi Shimmyo.

**Formal analysis:** Satoshi Okazaki, Shuken Boku, Ikuo Otsuka.

**Funding acquisition:** Satoshi Okazaki, Shuken Boku.

**Investigation:** Satoshi Okazaki, Shuken Boku, Yuichiro Watanabe, Takaki Tanifuji.

**Methodology:** Satoshi Okazaki, Shuken Boku, Ikuo Otsuka.

**Project administration:** Shuken Boku.

**Resources:** Satoshi Okazaki, Shuken Boku, Yuichiro Watanabe, Ikuo Otsuka, Tadasu Horai, Ryo Morikawa, Atsushi Kimura, Naofumi Shimmyo, Takaki Tanifuji, Toshiyuki Someya, Akitoyo Hishimoto.

**Supervision:** Shuken Boku, Toshiyuki Someya, Akitoyo Hishimoto.

**Validation:** Yuichiro Watanabe, Ryo Morikawa, Toshiyuki Someya.

**Writing – original draft:** Satoshi Okazaki.

**Writing – review & editing:** Shuken Boku, Akitoyo Hishimoto.

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
