## [Decision Letter · Decision Letter 0]

10 Jan 2022

PONE-D-21-30525Polymorphisms on the hypoxia inducible factor-binding site of the macrophage migration inhibitory factor gene promoter may be associated with the pathophysiology of schizophreniaPLOS ONE

Dear Dr. Boku,

Thank you for submitting your manuscript to PLOS ONE. After careful consideration, we feel that it has merit but does not fully meet PLOS ONE’s publication criteria as it currently stands. Therefore, we invite you to submit a revised version of the manuscript that addresses the points raised during the review process.

We look forward to receiving your revised manuscript.

Kind regards,

Md Ekhtear Hossain, Ph.D.

Academic Editor

PLOS ONE

Journal Requirements:

a) Did participants provide their written or verbal informed consent to participate in this study?

This work was partly supported by grants from Japan Society of the Promotion of Science (JSPS) KAKENHI grant numbers 15K19727, 18K15483, and 21K07520 (SO), 15K09805 and 18K07556 (SB), as well as 17H04249 (AH). 

This work was partly supported by grants from JSPS KAKENHI grant numbers 15K19727, 18K15483, and 21K07520 (SO), 15K09805 and 18K07556 (SB), as well as 17H04249 (AH). 

This work was partly supported by grants from Japan Society of the Promotion of Science (JSPS) KAKENHI grant numbers 15K19727, 18K15483, and 21K07520 (SO), 15K09805 and 18K07556 (SB), as well as 17H04249 (AH). 

Reviewers' comments:

Reviewer's Responses to Questions

**Comments to the Author**

1. Is the manuscript technically sound, and do the data support the conclusions?

Reviewer #1: Yes

Reviewer #2: Yes

2. Has the statistical analysis been performed appropriately and rigorously? 

Reviewer #1: Yes

Reviewer #2: Yes

3. Have the authors made all data underlying the findings in their manuscript fully available?

Reviewer #1: Yes

Reviewer #2: Yes

4. Is the manuscript presented in an intelligible fashion and written in standard English?

Reviewer #1: Yes

Reviewer #2: Yes

5. Review Comments to the Author

Reviewer #1: Comments to Authors

With interest, I read the manuscript by Satoshi Okazaki and colleagues. The authors investigated the involvement of hypoxia-induced MIF in the pathophysiology of schizophrenia (SCZ).

The authors are to be applauded to demonstrate that the introduction of SCZ-associated SNPrs 17004038 (C->A variant) on hypoxia-responsive element (HRE) region of MIF promoter significantly diminished hypoxia-induced MIF expression in primary astrocytes isolated from neonatal mice. Thus, the study serves as a reverse translation from bedside to bench in hypoxia-associated schizophrenia research.

The manuscript is well-written and straightforward.

However, the manuscript needs greater clarity to strengthen the findings.

Major comments:

1. The involvement of MIF in SZC was confused.

The current study concluded that SNP rs17004038 was significantly associated with SCZ. However, SNP rs17004038 (C—>A variant) significantly hampered hypoxia-induced MIF expression, suggesting that low levels of hypoxia-induced MIF are associated with SCZ.

However, the study also stated that elevated MIF levels were significantly associated with SCZ. and demonstrated that hypoxia promoted MIF expression. Further, perinatal hypoxia more likely resulted in SCZ development in later life, suggesting elevated levels of hypoxia-induced MIF are associated with SCZ.

Would you please clarify this?

2. The authors employed a meta-analysis to evaluate the association between SNPrs 17004038 and SCZ based on data from different set of participants. This is one of the key findings of the study. Yet, there is a significant lack of clarification of how the meta-analysis were conducted. For example, what was the weigth method used? How was 95%CI of combined Odds ratio calculated?

While the fixed-effect model was justified by Cochrane’s Q and significance were evidenced by a small p-value, the p-value is not an effect size. Because the pooled sample size in the meta-analysis was very large, the combined effect will be more likely significant, even if the combined effect is small. Therefore, estimated effect size (or Odds ratio) and its precision (indicated by the width of the confidence interval) should be interpreted, rather than the p-value. The combined Odds ratio was 1.46 (0.14-14.82) using the Meta-essentials package.

https://www.erim.eur.nl/research-support/meta-essentials/

Please clarify how was 95%CI of the combined Odds ratio was calculated?

3. Please clarify why two sets of participants within the same study (same study design, same study hypothesis, and data collection techniques and methods) were separated and analyzed independently before meta-analysis. If these two sets of participants had been combined as one, would the meta-analysis be necessary?

4. The sex dependency of the development of SCZ has been reported. In the current study, and sex distribution was significantly unequal between control and SCZ groups in 1st set of participants. Therefore, the analysis of 1st set of participants can be confounded by sex which were not controlled in the current study.

5. Although the authors stated to investigate the involvement of MIF in the pathophysiology of SCZ while focusing on the HIF pathway, there were no data on HIF expression under hypoxia conditions. Is the expression of HIFα/β altered in hypoxia condition (vs. normoxia)?

Additional comments:

6. Also, in line 11, the study did not study or present any data related to the “pathophysiology” of SCZ. Thus, please rephrase or rewrite the statement.

7. Also, please clarify/provide more information about the protective or pathological role of MIF in SCZ.

8. The title was too vague, line 2:“may be associated.”

Reviewer #2: The manuscript describes the role of Macrophage migration inhibitory factor (MIF), a cytokine that facilitates neurogenesis and neuroprotection, in the pathophysiology of schizophrenia (SCZ) with a focus on the hypoxia-inducible factor (HIF) pathway. The authors have demonstrated that single nucleotide polymorphisms (SNPs) on hypoxia response element (HRE) in the promoter region of MIF disturbed hypoxia-induced MIF expression and is associated with SCZ. The great importance of this manuscript is that this study elucidates the involvement of hypoxia in the potential pathophysiology of SCZ. Therefore, this work should be published to support this expanding field.

Few minor comments:

1. The difference between the wild type and SNP rs17004038 I.e., C>A is mentioned suddenly in the result section. Please include it in the abstract.

2. Page 6. Line 9. Ethics Statement. ‘All efforts were done to minimize the suffering of mice’. Please consider changing it to ‘were made’.

3. Page 6. Line 12. Please include the number of participants (Both control and SCZ patients)

4. A theoretical picture linking SNP on HRE in MIF promoter and MIF expression and association with SCZ would be helpful as a summary.

6. PLOS authors have the option to publish the peer review history of their article (what does this mean?). If published, this will include your full peer review and any attached files.

---

## [Author Response · Author response to Decision Letter 0]

20 Feb 2022

Reviewer #1: Comments to Authors

With interest, I read the manuscript by Satoshi Okazaki and colleagues. The authors investigated the involvement of hypoxia-induced MIF in the pathophysiology of schizophrenia(SCZ).

The authors are to be applauded to demonstrate that the introduction of SCZ-associated SNPrs 17004038 (C->A variant) on hypoxia-responsive element (HRE) region of MIFpromoter significantly diminished hypoxia-induced MIF expression in primary astrocytes isolated from neonatal mice. Thus, the study serves as a reverse translation from bedsideto bench in hypoxia-associated schizophrenia research.

The manuscript is well-written and straightforward.

However, the manuscript needs greater clarity to strengthen the findings.

Major comments:

1. The involvement of MIF in SZC was confused.

The current study concluded that SNP rs17004038 was significantly associated with SCZ. However, SNP rs17004038 (C—>A variant) significantly hampered hypoxia-inducedMIF expression, suggesting that low levels of hypoxia-induced MIF are associated with SCZ.

However, the study also stated that elevated MIF levels were significantly associated with SCZ, and demonstrated that hypoxia promoted MIF expression. Further, perinatal hypoxia more likely resulted in SCZ development in later life, suggesting elevated levels of hypoxia-induced MIF are associated with SCZ.

Would you please clarify this?

Response: Thank you very much for your invaluable comments. Previous studies have shown that blood MIF is increased in patients with SCZ, including initial-onset, drug-naive patients [32, 33]. Our previous studies showed that antipsychotic dosage positively correlated with MIF blood levels, that a higher-expression polymorphism of the MIF promoter is significantly lower frequent in female patients with adolescent-onset SCZ [34], and that clozapine and other antipsychotics increased MIF expression in PCAs [35]. These findings suggest that MIF may function as a physiological protective factor against SCZ risk [35]. The present study showed that SNP rs17004038 in the HRE of the MIF gene is significantly more common in patients with SCZ, and that this SNP inhibits hypoxia-induced MIF expression in PCAs from mice. Taken together, these studies suggest that elevated levels of hypoxia-induced MIF may have a protective function against perinatal hypoxia-associated SCZ development, and that SNP rs17004038 may inhibit the protective function of MIF by blocking MIF expression in response to hypoxia. We revised the Discussion as follows:

- Previous studies have shown that blood MIF is increased in patients with SCZ, including initial-onset, drug-naive patients [32, 33]. Our previous studies showed that antipsychotic dosage positively correlated with MIF blood levels, that a higher-expression polymorphism of the MIF promoter is significantly lower frequent in female patients with adolescent-onset SCZ [34], and that clozapine and other antipsychotics increased MIF expression in PCAs [35]. These findings suggest that MIF may function as a physiological protective factor against SCZ risk [35]. The present study showed that SNP rs17004038 in the HRE of the MIF gene is significantly more common in patients with SCZ, and that this SNP inhibits hypoxia-induced MIF expression in PCAs from mice. Taken together, these studies suggest that elevated levels of hypoxia-induced MIF may have a protective function against perinatal hypoxia-associated SCZ development, and that SNP rs17004038 may inhibit the protective function of MIF by blocking MIF expression in response to hypoxia. MIF may play a role in the pathophysiology of SCZ through both genetic factors (i.e., the SNP) and environmental factors (i.e., hypoxia), indicating that MIF may mediate the gene-environment interaction underlying the pathophysiology of SCZ.

(Page 25)

2. The authors employed a meta-analysis to evaluate the association between SNP rs 17004038 and SCZ based on data from different set of participants. This is one of the keyfindings of the study. Yet, there is a significant lack of clarification of how the meta-analysis were conducted. For example, what was the weigth method used? How was 95%CI ofcombined Odds ratio calculated?

While the fixed-effect model was justified by Cochrane’s Q and significance were evidenced by a small p-value, the p-value is not an effect size. Because the pooled sample sizein the meta-analysis was very large, the combined effect will be more likely significant, even if the combined effect is small. Therefore, estimated effect size (or Odds ratio) and itsprecision (indicated by the width of the confidence interval) should be interpreted, rather than the p-value. The combined Odds ratio was 1.46 (0.14-14.82) using the Meta-essentials package.

https://www.erim.eur.nl/research-support/meta-essentials/

Please clarify how was 95%CI of the combined Odds ratio was calculated?

Response: Thank you for your valuable comment. Referring to your point 3, we performed an analysis using the combined sample of the 1st and 2nd sets　instead of meta-analysis.

3. Please clarify why two sets of participants within the same study (same study design, same study hypothesis, and data collection techniques and methods) were separated and analyzed independently before meta-analysis. If these two sets of participants had been combined as one, would the meta-analysis be necessary?

Response: Thank you for your excellent suggestion. The 1st and 2nd sets were sampled from different areas of Japan, but as you pointed out, the two sets of participants were analyzed within the same study. Following your recommendation, we combined these two sets into one analysis, instead of doing a meta-analysis. We revised Table 1, added S3 and S4 Table, and revised the manuscript as follows:

- SNP rs17004038 was significantly associated with SCZ (p = 0.0424, odds ratio = 1.445),

(Page 2)

- The distribution of SNP rs17004038 (C>A) in patients with schizophrenia and controls was investigated. We found a non-significant trend for the association of SNP rs17004038 with SCZ in the first set of subjects (p = 0.0509) (S3 Table), but not in the second set (p = 0.364) (S4 Table). Next, we performed an analysis using both sets combined and found a significant association of SNP rs17004038 with SCZ (p = 0.0424). The odds ratio (1.445) suggests that the minor A allele of SNP rs17004038 may be involved in the pathophysiology of SCZ (Table 1).

(Page 15-16)

4. The sex dependency of the development of SCZ has been reported. In the current study, and sex distribution was significantly unequal between control and SCZ groups in 1st set of participants. Therefore, the analysis of 1st set of participants can be confounded by sex which were not controlled in the current study.

Response: Thank you for your valuable comment. As you pointed out, sex distribution was significantly unequal between control and SCZ groups in the first set of participants. Therefore, we additionally performed subgroup analysis divided by sex. As a result, each first and second set of subjects showed no significant difference in the frequency of the allele between SCZ and control (S3 and S4 Table). Combining the sets revealed a significant difference in the samples from female subjects (p = 0.0478) but not male subjects (p = 0.368), consistent with the overall analysis (Table 1). We revised the manuscript as follows:

- We additionally performed subgroup analyses divided by sex. Each set of subjects showed no significant difference in the frequency of the allele between SCZ and control (S3 and S4 Table). Combining the sets revealed a significant difference in the samples from female subjects (p = 0.0478) but not male subjects (p = 0.368), consistent with the overall analysis (Table 1).

(Page 16)

- First, we investigated the SNP rs17004038 (C>A) in the HRE of the MIF promoter. We found that rs17004038 is significantly associated with SCZ. The odds ratio indicates the minor A allele of rs17004038 is a risk allele for SCZ. Sex distribution was significantly unequal between control and SCZ groups in the first set of participants. Thus, we additionally performed subgroup analyses divided by sex. The samples from female subjects showed a significant difference, consistent with overall results, but the samples from male subjects did not. One reason is that the sample size of male subjects was relatively small, considering that the same minor allele frequency was present in both samples of male (SCZ: 0.0227 vs. CTL: 0.0183) and female (SCZ: 0.0240 vs. CTL: 0.0143) subjects, and that the samples from male subjects had smaller sample power (0.141) than that of female subjects (0.509). Another reason may be the influence of sex difference in the development of SCZ. Further studies with larger samples are required.

(Page 23)

5. Although the authors stated to investigate the involvement of MIF in the pathophysiology of SCZ while focusing on the HIF pathway, there were no data on HIF expression under hypoxia conditions. Is the expression of HIFα/β altered in hypoxia condition (vs. normoxia)?

Response: Thank you for your excellent suggestion. We investigated HIF-1α/β mRNA expression and HIF-1α protein level in the cell lysates of PCAs. We confirmed that hypoxia increased HIF-1α protein level in the cell lysates of PCAs, although HIF-1α/β mRNA levels were little changed. Under normoxic conditions, the intracellular concentration of HIF-1α protein is low, not due to lower protein expression but because it is negatively regulated by proteolysis by the ubiquitin-proteasome system via the von Hippel–Lindau tumor suppressor. Under hypoxic conditions, HIF-1α levels are stable and it translocates to the cell nucleus and associates with HIF-1β, and the HIF-1α/1β complex binds to the HRE of the DNA resulting in the transcription of various genes [46-48]. Our experimental system using PCAs under hypoxia seems to be in agreement with the results of previous studies. We added S1 Fig. and renamed the original S1 Fig to S2 Fig. We added the qRT-PCR primers of HIF-1α/1β in the S2 Table. We revised the manuscript as follows:

- HIF-1α protein concentration was measured with the Mouse HIF-1-alpha SimpleStep ELISA Kit (Abcam, Cambridge, UK) according to the manufacturer’s protocol.

(Page 11) 

- Effects of hypoxia on mouse HIF mRNA expression and HIF protein level

First, the effect of hypoxia on HIF-1α/β mRNA expression was examined. The HIF-1α/β mRNA expression levels were slightly changed by hypoxia, although significant changes were observed in HIF-1α mRNA level at 48 h and HIF-1β mRNA level at 24 h (S1 Fig A and B). Next, the effect of hypoxia on HIF-1α protein level was examined with ELISA. Hypoxic treatments for 24 and 48 h increased HIF-1α protein level in the cell lysates (S1 Fig C).

(Page 18)

- First, we confirmed that hypoxia increased HIF-1α protein level in the cell lysates of PCAs, although HIF-1α/β mRNA levels were little changed. Under normoxic conditions, the intracellular concentration of HIF-1α protein is low, not due to lower protein expression but because it is negatively regulated by proteolysis by the ubiquitin-proteasome system via the von Hippel–Lindau tumor suppressor. Under hypoxic conditions, HIF-1α levels are stable and it translocates to the cell nucleus and associates with HIF-1β, and the HIF-1α/1β complex binds to the HRE of the DNA resulting in the transcription of various genes [46-48]. Our experimental system using PCAs under hypoxia seems to be in agreement with the results of previous studies.

(Page 23-24)

- S1 Fig. Hypoxia induced hypoxia inducible factor (HIF)-1α/β mRNA expression and protein level. A The time-dependent effects of hypoxia on HIF-1α mRNA expression in primary cultured astrocytes (PCAs). PCAs were incubated for 3, 6, 12, 24, or 48 h in 21.0% or 0.1% O2. HIF-1α mRNA expression was analyzed by qRT-PCR. The values are shown as the ratio of HIF-1α mRNA to beta-actin (ACTB) mRNA (Student’s t-test; n = 6). B The time-dependent effects of hypoxia on HIF-1β mRNA expression in PCAs. PCAs were incubated for 3, 6, 12, 24, or 48 h in 21.0% or 0.1% O2. HIF-1β mRNA expression was analyzed by qRT-PCR. The values are shown as the ratio of HIF-1β mRNA to ACTB mRNA (Student’s t-test; n = 6). C The effects of hypoxia on HIF-1α protein level in cell lysate of PCAs. PCAs were incubated for 24 or 48 h in 21.0% or 0.1% O2. The HIF-1α protein level per 1 µg of total protein was analyzed using a HIF-1α enzyme-linked immunosorbent assay (ELISA) (Student’s t-test; n = 6). The data are expressed as the mean ± SEM. *p < 0.05, **p < 0.01, or ***p < 0.001.

(Page 44)

Additional comments:

6. Also, in line 11, the study did not study or present any data related to the “pathophysiology” of SCZ. Thus, please rephrase or rewrite the statement.

Response: Thank you for your valuable comment. We revised the Abstract as follows:

- We investigated the involvement of MIF in SCZ while focusing on the HIF pathway.

(Page 2)

7. Also, please clarify/provide more information about the protective or pathological role of MIF in SCZ.

Response: Thank you for your valuable comment. We clarified the protective or pathological role of MIF in SCZ, as your point 3 described above.

8. The title was too vague, line 2:“may be associated.”

Response: Thank you for your valuable comment. We revised the Title as follows:

- Polymorphisms in the hypoxia inducible factor binding site of the macrophage migration inhibitory factor gene promoter in schizophrenia

(Page 1)

Reviewer #2: 

The manuscript describes the role of Macrophage migration inhibitory factor (MIF), a cytokine that facilitates neurogenesis and neuroprotection, in thepathophysiology of schizophrenia (SCZ) with a focus on the hypoxia-inducible factor (HIF) pathway. The authors have demonstrated that single nucleotide polymorphisms (SNPs)on hypoxia response element (HRE) in the promoter region of MIF disturbed hypoxia-induced MIF expression and is associated with SCZ. The great importance of thismanuscript is that this study elucidates the involvement of hypoxia in the potential pathophysiology of SCZ. Therefore, this work should be published to support this expandingfield.

Few minor comments:

1. The difference between the wild type and SNP rs17004038 I.e., C>A is mentioned suddenly in the result section. Please include it in the abstract.

Response: Thank you for your valuable comment. We revised the Abstract as follows:

- First, we conducted an association study of the SNP rs17004038 (C>A) in the HRE of the MIF promoter between 1758 patients with SCZ and 1507 controls.

(Page 2)

2. Page 6. Line 9. Ethics Statement. ‘All efforts were done to minimize the suffering of mice’. Please consider changing it to ‘were made’.

Response: Thank you for your valuable comment. We revised the Ethics Statement as follows:

- All efforts were made to minimize suffering of mice.

(Page 7)

3. Page 6. Line 12. Please include the number of participants (Both control and SCZ patients)

Response: Thank you for your valuable comment. We revised the Pariticipants as follows:

- In the association study, all participants were of Japanese descent and recruited from the suburbs of Kobe city (first set, 915 patients and 836 controls) and Niigata city (second set, 843 patients and 671 controls) in Japan.

(Page 7)

4. A theoretical picture linking SNP on HRE in MIF promoter and MIF expression and association with SCZ would be helpful as a summary.

Response: Thank you for your excellent suggestion. We added Fig 4 and described putative roles of SNP rs17004038 on the HRE of the MIF gene promoter in perinatal hypoxia-associated schizophrenia (SCZ) risk as follows:

- Putative roles of SNP rs17004038 in the HRE of the MIF gene promoter in perinatal hypoxia-associated SCZ risk are shown in Fig 4.

(Page 25)

- Fig 4. Putative roles of single nucleotide polymorphism (SNP) rs17004038 in the hypoxia response element (HRE) of the macrophage migration inhibitory factor (MIF) gene promoter in perinatal hypoxia-associated schizophrenia (SCZ) risk. Abbreviation: HIF, Hypoxia-inducible factor.

(Page 26)

---

## [Editor Report · Decision Letter 1]

8 Mar 2022

Polymorphisms in the hypoxia inducible factor binding site of the macrophage migration inhibitory factor gene promoter in schizophrenia

PONE-D-21-30525R1

Dear Dr. Boku,

We’re pleased to inform you that your manuscript has been judged scientifically suitable for publication and will be formally accepted for publication once it meets all outstanding technical requirements.

Kind regards,

Md Ekhtear Hossain, Ph.D.

Academic Editor

PLOS ONE

---

## [Editor Report · Acceptance letter]

15 Mar 2022

PONE-D-21-30525R1 

Polymorphisms in the hypoxia inducible factor binding site of the macrophage migration inhibitory factor gene promoter in schizophrenia 

Dear Dr. Boku:

I'm pleased to inform you that your manuscript has been deemed suitable for publication in PLOS ONE. Congratulations! Your manuscript is now with our production department. 

Kind regards, 

on behalf of

Dr. Md Ekhtear Hossain 

Academic Editor

PLOS ONE